# Research on Multi-target Attack Scheme Based on K-means Clustering Algorithm

1st Zengzhi Chen
*Tianjin Jinhang Computing Technology Research Institute*
Tianjin, People's Republic of China
E-mail: hhjnczz@163.com

2nd Xiaoyu Song
*Tianjin Jinhang Computing Technology Research Institute*
Tianjin, People's Republic of China

3rd Yanling Li
*Tianjin Jinhang Computing Technology Research Institute*
Tianjin, People's Republic of China

*Abstract*—**The current battlefield environment is complex, with numerous and unevenly distributed targets in the attack area. It is difficult for commanders to make accurate real-time choices on the minimum amount of bombs to be dropped and the location of attack points. This article is based on the K-means clustering algorithm and analyzes multiple randomly distributed targets in the attack zone for two different combat situations. Mathematical models are established and corresponding attack schemes are studied. The experimental results show that the attack scheme proposed in this article can optimize the number of bombs dropped and provide accurate location information of each attack point, ultimately providing reliable attack schemes for commanders.**

*Index Terms*—**multi-target, amount of bombs dropped, attack point, K-means clustering algorithm**

## I. INTRODUCTION

Implementing precise strikes on multiple randomly distributed targets in the attack zone is one of the key factors determining the outcome of a war [1-4]. In order to reduce the probability of our fighter jets being detected and captured by the enemy, it is necessary to minimize the number of bomb drops and open positions as much as possible. At the same time, reasonable allocation of bomb deployment quantities can also save combat costs [5-7]. Therefore, to effectively destroy all enemy targets and reduce the consumption of bombs, it is necessary to plan the targets and the number of bombs dropped within the attack area under certain constraints [8-12]. This article is based on the K-means clustering algorithm [13-17] to study multi-target attack schemes. Finally, accurate, real-time, and efficient analysis was conducted on all target locations within the attack area. The research results can provide the minimum amount of bomb and attack point location for destroying all targets. Therefore, we can provide effective decision-making basis for battlefield deployment.

This article will study the following two types of combat situations:

(1) Given that there are a total of $n$ randomly distributed targets within a certain attack zone and their location information is known. The damage radius of bomb is $r$. If all targets within the attack area are destroyed, calculate the minimum number of bombs required and the location of each bomb's attack point.

(2) Given that there are $n$ randomly distributed targets in a certain attack area and their location information is known. The damage radius of bomb is $r$. The carrier carries a total of $m$ bombs (less than the minimum number of bombs required to destroy all targets in the attack area), calculate the attack point position of each bomb to maximize the damage effect, and calculate the damage rate.

## II. PRELIMINARIES

K-means clustering algorithm, as a representative of unsupervised clustering algorithms, is an iterative clustering analysis algorithm. The main function of this algorithm is to automatically classify similar samples into one category. The main idea of K-means clustering algorithm is to classify each data in the sample dataset into the class represented by the nearest cluster center when we give $k$ initial cluster centers. After all the data in the sample dataset is allocated, the cluster centers of each class are recalculated, and then the steps of allocating data and updating cluster centers are iteratively performed until the change in cluster centers is minimal or the specified number of iterations is reached.

The specific implementation process of the algorithm is as follows:

(1) Randomly select $k$ sample points from $n$ sample dataset as initial clustering centers $C_i(i = 1, 2, 3 \cdots k)$;

(2) For each sample in the sample dataset $a_j(x_j, y_j)(j = 1, 2, 3 \cdots n)$, Calculate its distance to $k$ cluster centers $d_j(j = 1, 2, 3 \cdots n)$. And classify it into the class corresponding to the cluster center with the smallest distance, where $d_j = \|a_j - C_i\|$;

(3) According to the previous allocation scheme, recalculate the class center for each class, which is the mean of all samples in that class $C_i = (\sum\limits_{a_j \in C_i} a_j)/p$, where $p$ is the number of sample data in each class;

Repeat steps (2) and (3) until the position of the cluster center no longer changes or reaches the set number of iterations, and output the clustering results.

## III. PROBLEM FORMULATION AND MAIN RESULTS

### A. Problem formulation

The distribution of targets on the battlefield is uneven. We can effectively use the K-means clustering algorithm to cluster the targets within the attack area, obtaining $k$ small attack areas with high aggregation and corresponding attack point positions. This enables commanders to accurately and quickly analyze the battlefield situation through the output results, and obtain reliable attack strategies, which is very meaningful for effectively striking targets within the attack area.

Assuming there are $n$ randomly distributed targets in the attack area, and the position information of each target is known as $a_j(x_j, y_j)(j = 1, 2, 3 \cdots n)$, The carrier carries $m$ types of air to ground bombs with a damage radius of $r$.

- *Problem 1*. If all $n$ targets in the attack area are destroyed, calculate the minimum number of bombs required $m$ and the attack point location of each bomb $C_i$;
- *Problem 2*. If the carrying capacity of the carrier is less than the minimum number of bombs required to destroy all targets in the attack area, calculate the attack point location of each bomb $C_i$ to maximize the destructive effect.

### B. Main results

For problem 1, based on the K-means clustering algorithm, we designed the process as shown in Fig.1. The specific implementation steps are as follows:

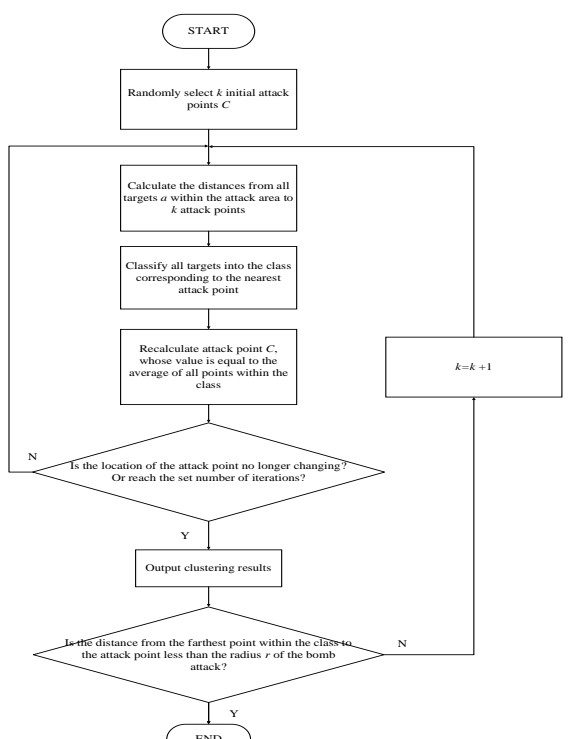

Fig. 1. Question 1 design flow chart.

Step 1: Let $k$=1, Randomly select one initial attack point within the attack zone $C_0(x_0, y_0)$;

Step 2: Calculate the distance $d_j$ from all target points $a_j(x_j, y_j)$ within the attack area to this attack point $C_0(x_0, y_0)$, where $d_j = \sqrt{(x_i - x_0)^2 + (y_i - y_0)^2}$;

Step 3: Classify all target points into the class corresponding to the nearest attack point (since there is only one attack point, all targets are classified into the class of the initial attack point $C_0$);

Step 4: Recalculate and generate new attack points $C_0(x_0, y_0)$, where $x_0$ is the average of the horizontal coordinates of all target points within the class, and $y_0$ is the average of the vertical coordinates of all target points within the class;

Step 5: Determine whether the attack point location has not been updated or has reached the set number of iterations. If so, proceed to step 6; If not, repeat step 2;

Step 6: Determine whether the distance between the target point within the class that is farthest from the attack point and the attack point is less than the bomb's damage radius. If so, end the process and output the attack scheme; If not, proceed to step 7;

Step 7: Let $k$=$k$+1, that is, increase the number of attack points by 1 each time, and repeat step 2.

Until all conditions are met, output the minimum required number of bombs $m$, which is the value of $k$, and the coordinates of the attack point, which is the coordinate of point $C$.

For problem 2, based on the K-means clustering algorithm, we designed the process as shown in Fig.2. The specific implementation steps are as follows:

Step 1: Let $k$=1 and randomly select one initial attack point $C_0(x_0, y_0)$ within the attack area;

Step 2: Calculate the distance $d_j$ from all target points $a_j(x_j, y_j)$ within the attack area to this attack point $C_0(x_0, y_0)$, where $d_j = \sqrt{(x_i - x_0)^2 + (y_i - y_0)^2}$;

Step 3: Classify all target points into the class corresponding to the nearest attack point (since there is only one attack point, all targets are classified into the class of the initial attack point $C_0$);

Step 4: Recalculate and generate new attack points $C_0(x_0, y_0)$ , where $x_0$ is the average of the horizontal coordinates of all target points within the class, and $y_0$ is the average of the vertical coordinates of all target points within the class;

Step 5: Determine whether the amount of ammunition $k$ used is greater than the amount of ammunition carried by the aircraft. If so, end the process and output the attack scheme; If not, let $k$=$k$+1 and repeat step 2.

Until all conditions are met, output the coordinates of the attack point and the damage rate.

## IV. NUMERICAL EXAMPLES

Based on the analysis of practical problems and mathematical models, we begin to conduct research on attack strategies for real-world battlefield problems. Assuming bomb

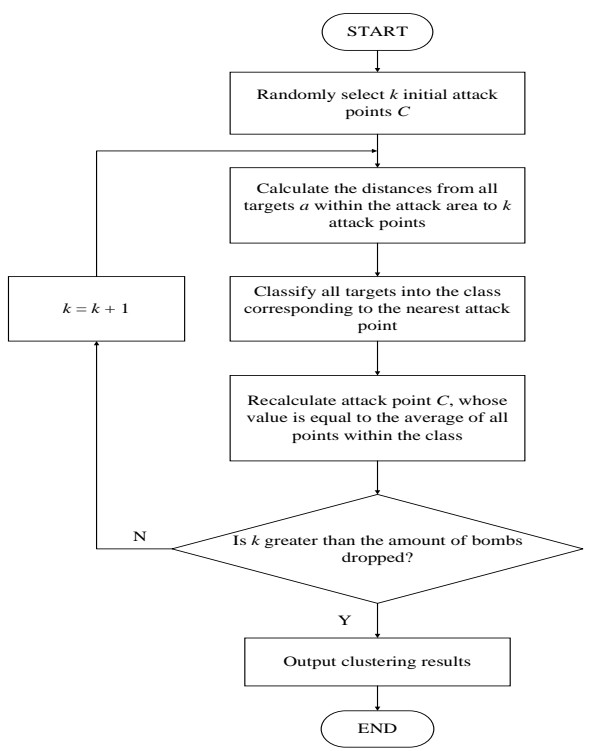

Fig. 2.   Question 2 design flow chart.

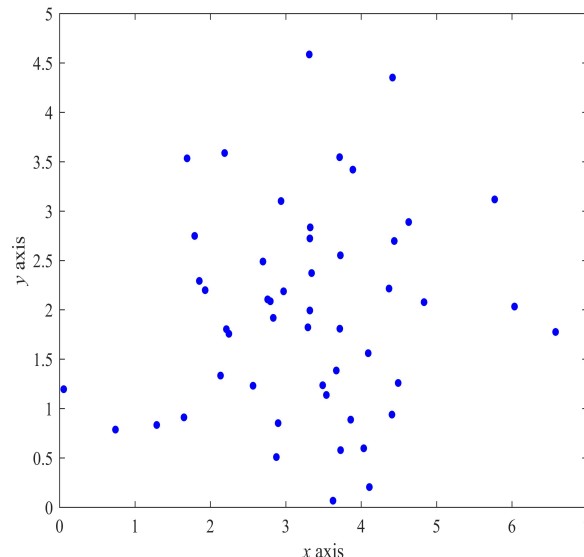

Fig. 3.   Randomly distributed target location.

within the attack area. The attack scheme is shown in Fig.4.

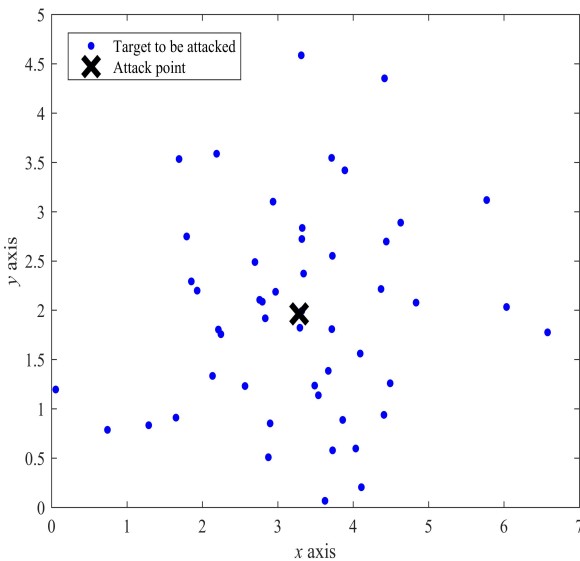

Fig. 4.   Attack scheme ($k$=1).

### TABLE I
#### RANDOMLY DISTRIBUTED TARGET LOCATION

| x | 3.5377 | 4.8339 | 0.7412 | 3.8622 | 3.3188 | 1.6923 | 2.5664 |
|---|--------|--------|--------|--------|--------|--------|--------|
| y | 1.1363 | 2.0774 | 0.7859 | 0.8865 | 1.9932 | 3.5326 | 1.2303 |
| x | 3.3426 | 6.5784 | 5.7694 | 1.6501 | 6.0349 | 3.7254 | 2.9369 |
| y | 2.3714 | 1.7744 | 3.1174 | 0.9109 | 2.0326 | 2.5525 | 3.1006 |
| x | 3.7147 | 2.7950 | 2.8759 | 4.4897 | 4.4090 | 4.4172 | 3.6715 |
| y | 3.5442 | 2.0859 | 0.5084 | 1.2577 | 0.9384 | 4.3505 | 1.3844 |
| x | 1.7925 | 3.7172 | 4.6302 | 3.4889 | 4.0347 | 3.7269 | 2.6966 |
| y | 2.7481 | 1.8076 | 2.8886 | 1.2352 | 0.5977 | 0.5776 | 2.4882 |
| x | 3.2939 | 2.2127 | 3.8884 | 1.8529 | 1.9311 | 2.1905 | 0.0557 |
| y | 1.8226 | 1.8039 | 3.4193 | 2.2916 | 2.1978 | 3.5877 | 1.1955 |
| x | 4.4384 | 3.3252 | 2.2451 | 4.3703 | 1.2885 | 2.8978 | 2.7586 |
| y | 2.6966 | 2.8351 | 1.7563 | 2.2157 | 0.8342 | 0.8520 | 2.1049 |
| x | 3.3192 | 3.3129 | 2.1351 | 2.9699 | 2.8351 | 3.6277 | 4.0933 |
| y | 2.7223 | 4.5855 | 1.3331 | 2.1873 | 1.9175 | 0.0670 | 1.5610 |
| x | 4.1093 | | | | | | |
| y | 0.2053 | | | | | | |

has a damage radius of 1.5 and 50 target points are randomly distributed in a certain combat area. The specific locations are shown in Table 1, and the distribution is shown in Fig.3.

Based on the analysis of the model established above, we will conduct the following research on this issue.

Firstly, we select one bomb to attack, as $k$=1. By incorporating the above model calculation, it can be concluded that the attack point is $C_1$=(3.2840, 1.9621), but the calculated maximum distance Dmax=3.3181 within the class (the distance from the farthest target point to the attack point within the class), which is more than the bomb's destructive radius of 1.5. Furthermore, it can be concluded that there are 21 target points outside the radius of 1.5, which can't destroy all targets

Increase the number of bullets used and attack with two bombs, as $k$=2. By incorporating the above model calculation, it can be concluded that the attack points are $C_1$=(4.1063, 2.0225) and $C_2$=(2.1486, 1.8787), but the calculated maximum distance Dmax=2.6829 within the class, which is more than the bomb's destructive radius of 1.5. Furthermore, it can be concluded that there are 13 target points outside the radius of 1.5, which can't destroy all targets within the attack area. The attack scheme is shown in Fig.5.

Increase the number of bullets used and attack with three

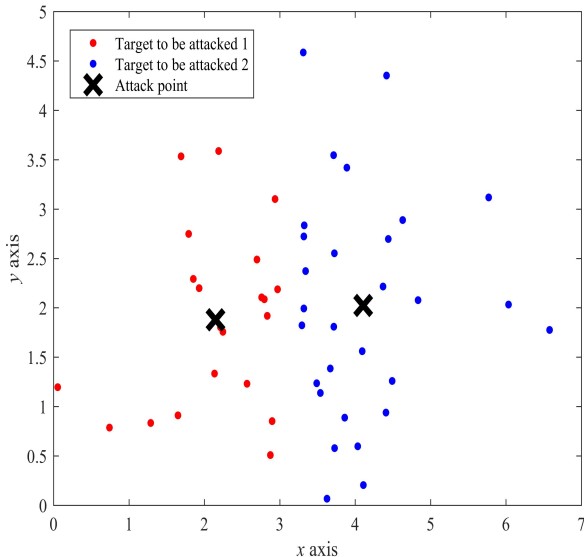

Fig. 5.    Attack scheme ($k$=2).

bombs, as $k$=3. By incorporating the above model calculation, it can be concluded that the attack points are $C_1$=(4.2899, 2.8927), $C_2$=(2.0227, 1.9440) and $C_3$=(3.6971, 1.0519), but the calculated maximum distance Dmax=2.5472 within the class, which is more than the bomb's destructive radius of 1.5. Furthermore, it can be concluded that there are 7 target points outside the radius of 1.5, which can't destroy all targets within the attack area. The attack scheme is shown in Fig.6.

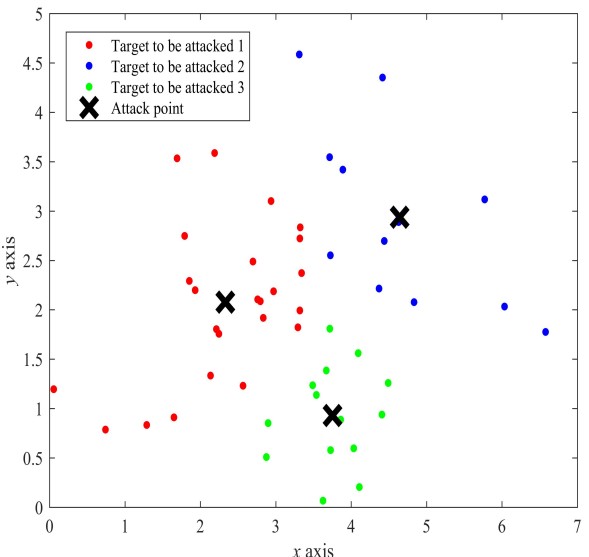

Fig. 6.    Attack scheme ($k$=3).

Increase the number of bullets used and attack with four bombs, as $k$=4. By incorporating the above model calculation, it can be concluded that the attack points

are $C_1$=(5.1341, 2.6441), $C_2$=(1.6679, 1.4340), $C_3$=(3.7224, 0.9892) and $C_4$=(2.9773, 2.8104), but the calculated maximum distance Dmax=1.8508 within the class, which is more than the bomb's destructive radius of 1.5. Furthermore, it can be concluded that there are 4 target points outside the radius of 1.5, which can't destroy all targets within the attack area. The attack scheme is shown in Fig.7.

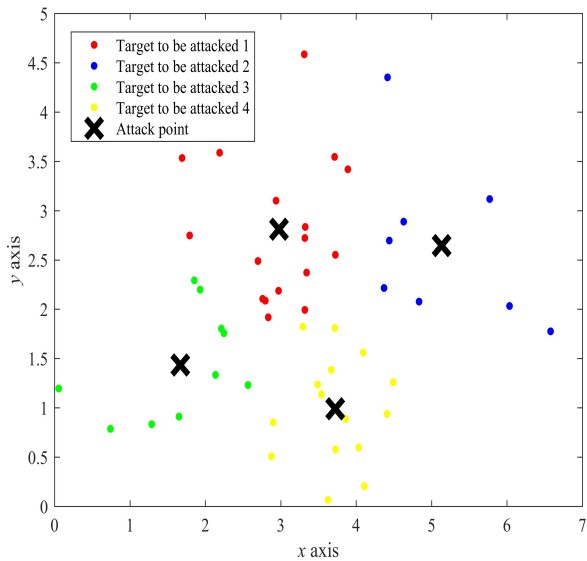

Fig. 7.    Attack scheme ($k$=4).

Increase the number of bullets used and attack with five bombs, as $k$=5. By incorporating the above model calculation, it can be concluded that the attack points re $C_1$=(5.2365, 2.4004), $C_2$=(3.1848, 3.6194), $C_3$=(0.9339, 0.9316), $C_4$=(3.7530, 0.9297) and $C_5$=(2.6936, 2.0945), and the calculated maximum distance Dmax=1.4950 within the class, which is less than the bomb's destructive radius of 1.5. Thus, all targets within the attack area can be destroyed. The attack scheme is shown in Fig.8.

In summary, if all 50 randomly distributed targets within the attack area are destroyed, a minimum of five bombs need to be used to attack, with the attack points being $C_1$=(5.2365, 2.4004), $C_2$=(3.1848, 3.6194), $C_3$=(0.9339, 0.9316), $C_4$=(3.7530, 0.9297) and $C_5$=(2.6936, 2.0945).

Additionally, we can conclude that when the carrier only uses one bomb to strike the attack area, the attack point should be chosen as $C_1$=(3.2840, 1.9621) to maximize the destructive effect, with a damage rate of 58%. When the carrier uses two bombs to strike the attack area, the attack point should be chosen as $C_1$=(4.1063, 2.0225) and $C_2$=(2.1486, 1.8787) to maximize the destructive effect, with a damage rate of 74%. When the carrier uses three bombs to strike the attack area, the attack point should be chosen as $C_1$=(4.2899, 2.8927), $C_2$=(2.0227, 1.9440) and $C_3$=(3.6971, 1.0519) to maximize the destructive effect, with a damage rate of 86%. When the carrier uses four bombs to strike the attack area, the attack

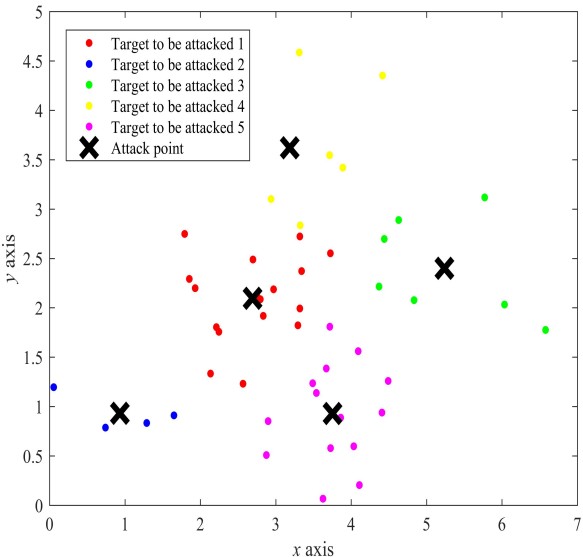

Fig. 8.    Attack scheme ($k$=5).

point should be chosen as $C_1$=(5.1341, 2.6441), $C_2$=(1.6679, 1.4340), $C_3$=(3.7224, 0.9892) and $C_4$=(2.9773, 2.8104) to maximize the destructive effect, with a damage rate of 92%.

## V. Conclusions

This article is based on the K-means clustering algorithm and studies the bomb deployment strategy for multi-target in the attack zone for two combat situations. The minimum number of bombs required to destroy all targets in the attack zone and the corresponding attack point coordinates are obtained, and the damage rate when using bombs with less than the minimum number of bombs for attack is also obtained. The experimental results show that the proposed attack scheme can accurately and reliably cluster all targets in the battlefield, which can provide effective decision-making basis for battlefield deployment.

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
