# OpenReview forum: "Research on Multi-target Attack Scheme Based on K-means Clustering Algorithm"
_IEEE.org/ICIST/2024/Conference — IEEE ICIST 2024 Conference Submission_

### Official Review · Reviewer_12Jx · 2024-08-30
**The research in this paper is interesting**

**Rating:** 8
**Confidence:** 5

**Review:**

1. Given the challenges posed by unevenly distributed targets, how does the proposed model adapt to varying densities of targets? Are there any specific strategies or modifications within the K-means algorithm that address this issue effectively?
2. Was a sensitivity analysis conducted to evaluate how changes in target distributions or the number of bombs affect the optimal attack schemes? How robust are the proposed solutions when faced with varying battlefield scenarios?

---

### Official Review · Reviewer_EWND · 2024-08-31
**This article is based on the K-means clustering algorithm and analyzes multiple randomly distributed targets in the attack zone for two different combat situations, and the corresponding attack schemes studied are reliable.**

**Rating:** 6
**Confidence:** 5

**Review:**

1.The rationale for using the K-means method instead of other techniques should be clearly outlined in the introduction. Moreover, it is advisable to add comparisons with alternative methods in the simulation section to emphasize the strengths of this research.
2.How do the authors handle initialization, outlier processing, and the automatic determination of K in the K-means clustering method presented for the scenario in this paper?
3.The font size in Figure 1 is too small. It is recommended to readjust the size for better readability.
4.The formatting of the references needs to be carefully reviewed and adjusted.

---

### Official Review · Reviewer_esxu · 2024-09-03
**Research on Multi-target Attack Scheme Based on K-means Clustering Algorithm**

**Rating:** 7
**Confidence:** 4

**Review:**

1.The paper is well-structured and clearly outlines the research problem, methodology, and results. The introduction section effectively sets the context and motivation for the study. However, some minor improvements in the clarity of explanations, especially in the problem formulation and main results sections, would further enhance the readability for a broader audience. Consider adding more details or examples to illustrate the concepts and steps involved in the proposed approach.

2.The experimental results demonstrate the effectiveness of the proposed attack scheme based on the K-means clustering algorithm. However, a more comprehensive discussion of the results would strengthen the paper. It would be beneficial to analyze the sensitivity of the results to different parameters, such as the number of targets, bomb damage radius, and the distribution of targets. Additionally, a comparison with other relevant algorithms or approaches would provide valuable insights into the advantages and limitations of the proposed method. Overall, a deeper analysis and discussion of the experimental findings would make the conclusions more convincing and impactful.

---

### Decision · Program_Chairs · 2024-09-06

Accept (Oral)